# Geochemical Characteristics of the Vertical Distribution of Heavy Metals in the Hummocky Peatlands of the Cryolithozone

**DOI:** 10.3390/ijerph20053847

**Published:** 2023-02-21

**Authors:** Roman Vasilevich, Mariya Vasilevich, Evgeny Lodygin, Evgeny Abakumov

**Affiliations:** 1Institute of Biology, Komi Science Center, Ural Branch, Russian Academy of Sciences, 167982 Syktyvkar, Russia; 2Department of Applied Ecology, Faculty of Biology, Saint Petersburg State University, 199178 St. Petersburg, Russia

**Keywords:** subarctic region, peatlands, histosols, biogeochemical barriers, ecotoxicants, technogenesis

## Abstract

One of the main reservoirs depositing various classes of pollutants in high latitude regions are wetland ecosystems. Climate warming trends result in the degradation of permafrost in cryolitic peatlands, which exposes the hydrological network to risks of heavy metal (HM) ingress and its subsequent migration to the Arctic Ocean basin. The objectives included: (1) carrying out a quantitative analysis of the content of HMs and As across the profile of Histosols in background and technogenic landscapes of the Subarctic region, (2) evaluating the contribution of the anthropogenic impact to the accumulation of trace elements in the seasonally thawed layer (STL) of peat deposits, (3) discovering the effect of biogeochemical barriers on the vertical distribution of HMs and As. The analyses of elements were conducted by atom emission spectroscopy with inductively coupled plasma, atomic absorption spectroscopy and scanning electron microscopy with an energy-dispersive X-ray detecting. The study focused on the characteristics of the layer-by-layer accumulation of HMs and As in hummocky peatlands of the extreme northern taiga. It revealed the upper level of microelement accumulation to be associated with the STL as a result of aerogenic pollution. Specifically composed spheroidal microparticles found in the upper layer of peat may serve as indicators of the area polluted by power plants. The accumulation of water-soluble forms of most of the pollutants studied on the upper boundary of the permafrost layer (PL) is explained by the high mobility of elements in an acidic environment. In the STL, humic acids act as a significant sorption geochemical barrier for elements with a high stability constant value. In the PL, the accumulation of pollutants is associated with their sorption on aluminum-iron complexes and interaction with the sulfide barrier. A significant contribution of biogenic element accumulation was shown by statistical analysis.

## 1. Introduction

Globally, wetland ecosystems act as the biosphere’s “lungs”, as they control the atmospheric pool [1,2]. The weak desalination of nonorganic components with atmospheric precipitation is known to ensure their good preservation in peat horizons [3,4]. A layer-by-layer analysis of the peat microelement composition offers the possibility of reconstructing the content of elements in the atmosphere over an extended period of time [1,5,6,7].

The subarctic region of the European Northeast is especially sensitive to climate change [8]. The soils of peat hummocks within the seasonally thawed layer (STL) are characterized by predominantly positive or high negative average annual temperatures (−1…+2 °C), thus being the most temperature-vulnerable “segment” among the permafrost soils of the East European Plain [9,10]. As a result of large-scale degradation of the permafrost in the cryolithozone, thawing organic deposits become involved in the carbon cycle as well [11]. This is also be facilitated by such co-occurring geomorphological processes as thermokarst and erosion, terminating in the exposure of deep deposits of organic matter [12].

However, at the same time, triggered by a changing climate, peat deposits containing significant amounts of bioavailable heavy metals (HMs) turn into sources of the toxicant migration into the environment [13,14,15]. Climate change and anthropogenic pressure caused by the intensive development of the oil and gas industry in the Arctic region result in a changed hydrological regime of the Arctic wetlands and the irreversible disturbance of peat layers. These processes will involve HMs conserved in the deep layers of peat in the cycle. Destabilized peatlands in the Arctic permafrost are a global environmental threat of inorganic pollutants entering the hydrological network and subsequently migrating to the Arctic Ocean basin [16,17,18].

Monitoring of the trace element content in peat deposits is often used to diagnose pollution around large industrial centers [19]. The most adverse impact on the Arctic areas of the European Northeast is known to be exerted by the coal and heat power industries. The Vorkuta industrial agglomeration located in the tundra zone displayed contamination, mainly with pulverized-fuel ash, which is distributed within a radius of over 30 km [20,21]. Similar conclusions were obtained for the area of Inta, where elevated concentrations of Al, Ba, Ca, K, Mg and Sr in snow cover and organogenic horizons of mineral soils were observed [22]. A similar diagnosis of Histosols in the Subarctic region of the European Northeast has not been previously performed.

Despite the foregoing, some researchers point to the uneven (often abnormal) accumulation of quite a number of elements along the profile, associated both with the migration and deposition characteristics of the elements, with the botanical composition of peat, the hydrological characteristics of peatland massifs, and the mineralogical composition of the soil-forming material [23]. Different taxonomic groups of wetland vegetation accumulate particular substances from the environment in different ways, which affects the chemical composition and properties of peat layers [24].

The objectives of this study were (1) carrying out a quantitative analysis of the content of acid- and water-soluble forms of HMs and As across the profile of Histosols in background and technogenic landscapes of the Subarctic region, (2) evaluating the contribution of the anthropogenic impact of the energy plants to the accumulation of trace elements in the STL of peat deposits, and (3) discovering the effect of biogeochemical barriers (sorption, cryogenic and biogenic ones) on the vertical distribution of HMs and As.

## 2. Materials

The area under study is located in the extreme northern taiga subzone of the European Northeast of Russia (the basin of the Kosyu River, part of the Pechora River basin), on the territory of sparsely insular permafrost [9]. The relief is represented by gently sloping moraine plains of the Cis-Ural region, covered with mantle silty loams with a thickness of below 10 m. The climate is temperate continental, the average annual air temperature at the weather station of Inta is minus 3.9 °C, the average daily sum of positive temperatures is about 1400 °C, and the average annual precipitation is approximately 700 mm [25].

The entire thickness of peat in the soils of peat hummocks as part of hummocky-hollow peatland complexes, including permafrost layers (PL), was the object of our study. Sampling was carried out at two sites confined to two peatland areas. Plots 1 and 2 are deposited on a site seen as conditionally background (65°54′05.0′′ N 60°26′34.5′′ E) 20 km south-east from Inta. Layer-by-layer collection of samples was performed to a depth of 2.6 m (Plot 1) and 1.8 m (Plot 2). Plot 3 was selected at a site 5 km north-west from Inta (66° 05′ 05.4′′ N 59° 58′ 36.3′′ E) in order to assess the degree of aerotechnogenic effects of local emission sources (the Inta coal heat power plant and coal mines) (Figure 1). Layer-by-layer collection of samples was performed to a depth of 2.5 m. Based on the keys of the World Reference Base for Soil Resources [26], peatland soils are classified as Fibric Folic Cryic Histosols—soils under vegetation (Plots 1 and 3)—and the soils of eroded peat patches (Plot 2) are Hemic Folic Cryic Histosol (Turbic).

The vegetation cover on the hummocks is composed of shrub-lichen communities, mainly of lichens (g. *Cladina*), *Empetrum*, *Vaccinium vitis-idaea*, *V. uliginosum*, *Rubus chamaemorus*, *Betula nana* at the hummock edge, green mosses (gg. *Dicranum*, *Polytricum*). The edge of the site on an eroded peat patch (Plot 2) is covered with lichens and green mosses. The upper limit of permafrost in summer is at a depth of 40–50 cm in soils under vegetation and 60–70 cm in soils with eroded cover. Peat in the upper and lower parts of the profiles is dark brown, with a high or medium degree of decomposition, belonging to the humus type, displaying a low or medium degree of decomposition in the central part.

## 3. Methods

Peat samples were collected every 5–10 cm. A sample containing mixed vegetation material (combing) consisting of dominant species listed above was obtained directly from the 50 × 50 cm surface of Plot 1. The chemical analysis of peats was performed at the environmental analytical laboratory (Institute of Biology, Komi RC Ur. Br. RAS). The analyses of acid-soluble forms of macroelements (Na, K, Mg, Ca, Al and Fe) and HMs (Hg, Pb, Cd, Cu, Zn, Cr, V, Ni, Mn, Co, Sr and Ba), S, P, and As, water-soluble forms and exchangeable Ca and Mg ions (Σ (Ca^2+^ + Mg^2+^)exch.) were conducted by atom emission spectroscopy with inductively coupled plasma using Spectro Arcos and Spectro Cyros CCD instruments (Spectro Analytical Instruments GmbH, Kleve, Germany). The mass concentration of total Hg was determined by atomic absorption spectroscopy on an RA-915+ analyzer (Lumex, St. Petersburg, Russia). The mineralization of peat samples for the subsequent determination of acid-soluble forms of HMs was carried out on a Minotaur-2 microwave system (Lumex, St. Petersburg, Russia) using a mixture of HNO_3_ (c) and H_2_O_2_ (c) in a ratio of 10:1. The mass concentrations of total organic carbon (TOC) and total nitrogen in water extracts were performed using a Shimadzu TOC-V_CPN_ analyzer (Shimadzu Corporation, Kyoto, Japan). The morphological analysis of mineral particles was determined on a Tescan Vega 3 LMH (Tescan, Brno, Czech Republic) scanning electron microscope (SEM). Elemental compositions were performed by energy dispersive X-ray spectroscopy (EDS) using X-MAX 50 (Oxford instruments, Abingdon, Great Britain) at the Institute of Geology (Komi RC, Ur. Br., RAS). Humic (HAs) and fulvic acid (FAs) preparations were extracted from the peats according to the IHSS method [27]. The paleobotanical composition and peat decomposition degree (*R*) were examined at the laboratory of wetland ecosystems (Institute of Biology, Karelian RC, RAS). Radiocarbon (^14^C) dates of the peat layers were measured in the Common Use Center, “Laboratory of Radiocarbon Dating and Electron Microscopy”, of the Institute of Geography of the RAS (lab code IGAN, Moscow, Russia). Radiocarbon activity was measured on an ultralow-background liquid scintillation alpha/beta spectrometer Quantulus 1220 (PerkinElmer, Turku, Finland). Calibration was performed using a CALIB Rev7.1.0 radiocarbon calibration program. The error of the dating method did not exceed 70–80 years for each date [6,23,28]. The paleogeographic scale of the Holocene was considered following the Blytt–Sernander sequence, with a modified Holocene chronological standard for the tundra and forest zones of Northern Eurasia [7].

Bivariate correlation analyses were conducted using the Pearson product-moment correlation coefficient (*r*), and its statistical significance was assessed via the Neyman–Pearson approach (normal distribution). Technically, the observed value of the coefficient (based on *n* pairs) was compared against the critical value (*r*_cr_) for a two-tailed test and significance level (*p*) of 0.95. Principal component analysis (PCA), using Statistica v. 12.1 (Dell, Round Rock, TX, USA), was performed to determine the correlations between the content of toxicants, botanical composition, physical and chemical parameters of peat. The number of the factors extracted from the variables was determined by the Kaiser rule. With this criterion, the first two principal components with an eigenvalue greater than two were retained [29].

## 4. Results and Discussion

### 4.1. Botanical Composition and Peat Age

The dominant plants, type and degree of peat decomposition in the studied sections, as well as the radiocarbon dating values of peat, are given in Table 1. A detailed description of the paleobotanical stages and the chronology of the studied peatlands was published earlier [30].

According to radiocarbon dating, the initial stage of peat accumulation belongs to the late boreal period BO2 8953 cal year BP. The most intense accumulation of peat in the studied sections falls on the Atlantic period of the Holocene 8953–5661 cal years BP. The lower part of the profiles was formed under eutrophic conditions with the dominance of herbaceous-sedge communities (*Carex caespitosa*, *C. aquatilis*, *Equisetum fluviatile*, *Menyanthes trifoliata*) (65–90%) and sometimes well-decomposed woody (25–30%) remains (*Betula pubescens*, *Pinus sylvestris*). Plot 3 is characterized by a significantly smaller share of *Equisetum fluviatile* and the absence of *Carex caespitosa* in the botanical composition of the lower part of the profile. The peat is well decomposed. The following stage (7421 cal year BP) of transitional peat shows a higher share of *Menyanthes trifoliata*, with some less demanding mineral nutrition *Carex limosa*, *Eriophorum* sp., and mesotrophic sphagnum mosses.

A low degree of peat decomposition and the plant species indicate waterlogged conditions for the formation of a peat deposit. Starting from 5661 cal year BP, the vegetation cover began to be dominated by *Scheuchzeria palustris*, the remains of which predominate in the peats of Plots 1 and 2 up to the very surface. The peat profile of Plot 3 in its central part is dominated by mesoeutrophic species *Carex chordorrhiza*, *C. limosa*, *C. rotundata*, *Menyanthes trifoliata* and *Equisetum fluviatile*, as well as woody remains of *Betula* sp. The peat displays a high degree of decomposition (*R* = 30–40%). The upper 30 cm of the studied sections are predominantly represented by shrub-sphagnum and sphagnum (*Sphagnum russowii*, *Pleurozium*, *Polytrichum*, *Ericales* and *Ledum*) types of peat with a low degree of decomposition, which began to be deposited at the beginning of the Subatlantic period (2635 cal year BP). The start of the permafrost establishment in the studied area is likely to date back to 2100–2300 cal year BP [31].

### 4.2. Physical and Chemical Properties, Macroelement Composition

The STL peat samples are acidic with pH 3.4–4.0 up to the upper limit of permafrost (oligotrophic peat formation). The top part of the STL (0–40 cm) of peatlands shows a trend to increasing values of the specific electrical conductivity of the water extract up to 80 μSm/cm (Plot 1) and 170–185 μSm/cm (Plot 2). The concentration of both the total exchangeable (30–127 mmol/kg) and the total acid-soluble forms of calcium and magnesium (0.14–2.05%) is low in Plots 2 and 3; however, it increases downwards the profile (Figure 2). The increased content of exchangeable forms (81–98 mmol/kg) and the sum of gross forms of calcium and magnesium (1.7–1.8%) for the upper peat layers of Plot 1 are related to dust pollution from nearby gravel-sand pits and unpaved roads. The ash content of the STL peat of the conditionally background area is 2–6%, and the carbon content is 54–59%. The ash content of peat in the upper horizons of the site in the Inta agglomeration exposure zone (Plot 3) reaches 11%. The lower peat layers (eutrophic peat formation) of all sections are characterized by higher pH values (up to 5.2), which are associated with an increase down the profile of the sum of exchangeable bases (up to 356 mmol/kg) and an ash content of up to 14–17% (in the central part of the profile). The mass fractions of Fe, Al and the totals of Ca and Mg in the acid-soluble extract systematically increase down the profile, reaching their peaks of 2.4, 4.6 and 8.9%, respectively, in the soil-forming material.

The distribution of Eh values is characterized by conditions gradually changing from transient (from weakly oxidizing to weakly reducing) in the STL (310–447 mV) to moderately reducing (240–290 mV) in the PL below 80–100 cm deep, where reduced forms of iron and manganese should prevail. The highest concentration of HAs is found in the STL, as it reaches 21.1% and smoothly decreases down the profile (Figure 2).

### 4.3. Microelement Composition of Peat Deposits in Hummocky Peatlands

#### 4.3.1. Element Accumulation Levels in the Peatland Profiles

Quantitative data analysis made it possible to build series of HMs in peat horizons of hummocky peatlands against their average content in the Earth’s crust (Ki) [32] and to identify the levels of element accumulation (Figure 3). The upper level of microelement accumulation is associated with the STL of peatlands, thus reflecting the degree of aerogenic pollution over an extended period of time, and is connected with lifetime accumulation of Hg, Cd, Pb, Zn, and other HMs by plants and peat organic matter. The type of element distribution in the upper peat horizons is largely related to the element concentration in plant material (Figure 3A–C). However, in the upper layers of peat, a number of elements (Hg, Pb, Cd) are more concentrated than in plant material. The humification of plant residues results in the formation of HAs, the main HM binding sites [33,34,35]. The content of other HMs in the oligotrophic part of the peat profile is significantly lower than their average content in the Earth’s crust. Plot 3, located in the local pollution exposure zone (Figure 3F), is characterized by a more intense accumulation of such elements as Hg, Pb, Cd, As, Ni, Co, Cr and V in the upper peat layers in comparison with Plot 1 (Figure 3B).

The values of the total concentration factor of HMs relative to the average content in the Earth’s crust (ΣKi) decrease down the profile to the upper permafrost boundary (40 cm) from 8.7 to 4.4 for Plot 3 and from 6.8 to 2.5 for Plot 1 of the conditionally background area. It proves the HM accumulation in the surface layer of peat, both as a result of long-range atmospheric transport and due to exposure to local emission sources [21]. ΣKi gradually increases within the range of 2.3–18.0 down the profile below the permafrost boundary for Plots 1 and 2, with its highest point identified at a depth of 190–230 cm (Figure 3D); the increased ΣKi is largely caused by the excess of clarkes and hygienic standards for soils for As (up to 4.0 times) and Cd (up to 2.3 times). The accumulation of HMs in these horizons in contrast to the soil-forming material, which displays a uniform distribution of elements in reference to their average content in the Earth’s crust, is likely to be associated with the intensive biogenic accumulation of elements. Plot 3 is characterized by a significantly lower HM accumulation in permafrost horizons, with ΣKi 2.6–13.7 and no excess of hygienic standards for all elements, except for a significant excess of As (up to 6.7 times).

#### 4.3.2. Assessment of the Level of Anthropogenic Pollution in the Upper Peat Layers Based on the Analysis of the Morphology, Size and Composition of Microparticles

Peat samples from the conditionally background area and the site of the Inta power plant exposure were subject to geochemical sampling using SEM. The analysis showed the presence of polydisperse particles of different morphology and size. In accordance with the linear dimensions, the particles were distributed into three clusters: 10–50 µm (up to 200 µm at times), 2–10 µm, and less than 2 µm (Figure 4). The biggest non-organic particles range in size from 10 to 50 µm (up to 200 µm) and demonstrate irregular morphology. These particles consist of silicon oxide and aluminum oxide with an admixture of Ti, Fe, Ca, Mg, Na, K and P compounds (Figure 4B-4). The spread of soil erosion particles through dusting from easily eroded surfaces under the action of wind is one of the main sources of coarsely dispersed aerosols of natural origin [36]. Microphotographs revealed the highest occurrence of these particles in the peat surface layer of a conditional background area (Plots 1 and 2). As stated earlier, this is associated with the drift of coarse particles from sand pits developed near the studied site. The second cluster is represented by pelitic particles (2–10 µm) and composed of quartz and aluminosilicates with an admixture of Fe, Ca, Ti, Zr, Y, Sc, and occasionally occurring rare earth elements: La, Ce, Nd and Th. These particles originated as a result of erosion, as well from soil-forming and rock materials. Particles of transitional dimension (about 5 μm) in terms of their elemental composition (C, O, Si, Al, S, Fe, Ni, Zn and Cu) are likely to be attributable to anthropogenic reasons as a result of exposure to coal extracting (mainly ash dumps) and energy generating sites in the region under study [37].

The third cluster has particles of less than 2 μm in size (Figure 4A1) and is represented by spheroidal, spherical with a clear and relief surface, hemispherical, hollow and angular particles, which are based on aluminosilicates with an admixture of elements such as Ti, Ca, Mg, K, S, P, and frequently Zn, Mn, Ni, V, Cr and occasionally Pb, Hg and Cd.

The X-ray phase analysis showed that SiO_2_ (up to 60%), Al_2_O_3_ (up to 18%) and Fe_2_O_3_ (up to 9%) are the main components of the chemical composition of aluminosilicate globules of pulverized-fuel ash [38]. One may encounter spherical (1–2 μm) and angular particles, consisting mainly of iron oxides and hydroxides enriched in Zn, Cr, Ni, as well as large globules of amorphous aluminosilicate glass, the surface of which is encrusted with iron oxides (Figure 4A2,3). The microporous surface and significant sorption capacity of this group of particles determine the accumulation of HMs in pulverized-fuel ash [37]. Particles of this cluster found in peat are related to the surface of plant tissues, the surface of large mineral particles and large aggregates of terrigenous origin. This group of particles was only noted in the 5 cm subsurface oligotrophic layer of the peatland in the area exposed to aerogenic pollution (Plot 3). A sufficiently high density of these particles per area unit points to a significant technogenic impact on the studied area. Micronic and ultramicronic particles are known to create a substantial hazard to public health, causing respiratory diseases [39,40]. Pulverized-fuel ash of power plants operated on high-ash bituminous coals may serve as regional sources of particles of about 2 μm (microspheres), which are based on aluminosilicates and iron oxides and hydro-oxides [19,38,41]. The composition and properties of microspheres are related to the conditions of fuel combustion and the association of minerals in coal. The share of microspheres may vary <0.1 to 3% of the mass of fly ash, and depends on the nature of the coal and the concentration of mineral impurities that form the glass phase [42]. Spherical and spheroidal particles may have significant linear dimensions up to 20–30 μm (Figure 4A2,C). This indicates their ingress from a local source—the Inta power plant. Previously, we recorded the presence of such particles in the snow cover near the city of Vorkuta [43]. The propagation of particles of this size over sufficiently large distances can be associated with their hollow structure and high volatility due to significant windage. Some microphotographs revealed fragments of hollow particles (cenospheres) with a destroyed shell (less than 1 μm thick) and plerospheres containing smaller globules (Figure 4C and Figure 5). The resistance of these particles to erosion and dissolution under acidic and aerobic conditions of acrotelm and their unique morphology (spheroidal and spherical), size and chemical composition allow their use as indicators of high-temperature combustion of solid fuels and their ingress resulting from long-distance and regional atmospheric air-mass transport.

#### 4.3.3. Assessment of the Level of Anthropogenic Pollution Based on the Concentration of Microelements in the Upper Part of the Peat Profile

Among the most typical elements present in the upper peat horizons, the most toxic ones—Hg, Pb and Cd—are mainly of anthropogenic origin and pose a real environmental threat to the Arctic ecosystems [44,45]. The available data indicate the accumulation of these elements in the upper part of the Arctic peat profiles [4]. The comparison of data on the HM content in the upper horizons of Plot 3, when compared to background areas (Plots 1 and 2), showed the excess of the following chalcophile elements: Hg 1.2 times, Cd 1.6 times, Pb 2.0 times, Cu 3.1 times and As 1.2 times. According to existing models, the largest part of Hg, Cd and Pb reaches the Arctic by air [45,46]. It was previously determined that these elements (except for As) accumulate in the organogenic soil horizons of the background landscapes of the tundra zone to the north of the study area, and their highest concentrations are confined to soils with a high content of organic matter [20].

Siderophile elements, which act as markers of coal combustion at heat power plants, contribute to the pollution of the surface layers of peat in Plot 3 the most [21]. The excess of Ni by 1.8 times, Co by 2.1 times, Cr by 3.7 times, V by 3.3 times and Fe by 4.6 times was established as compared to the data of the conditionally background area. Coal combustion at power plants causes manifold excesses of concentration of such elements as Hg, Pb, V, Zn and Cr in the pulverized-fuel ash [41]. An assessment of the content of heavy metals in soils and soil-like bodies of Vorkuta revealed a manifold excess of hygienic standards for Pb, Zn, Hg, Cu, As and Mn [20,22]. Earlier, we demonstrated that the highest degree of pollution of the adjacent territory caused by heat power plants in the northern latitudes occurs within the 10 km distance as a result of the deposition of dust (ash) enriched in various elements, including heavy metals [21]. Thus, the accumulation of HM and As in the upper peat layers, both as a result of transboundary atmospheric transport and exposure to local and regional emission sources, is shown.

#### 4.3.4. Analysis of the Profile Distribution of Elements

The distribution of microelements across the peat layers of hummocky peatlands is conditioned by the effect of geochemical barriers. Due to the low share of mineral components in the composition of raised-bog peat, the effect of the sorption barrier in the STL is mainly associated with the formation of HM humates and fulvates. The second main mechanism of their accumulation is in vivo absorption by peat-forming plants and adsorption by the surface of plant tissues. The rather high chemical affinity of Hg, Pb and Cd with HAs causes their intense accumulation in subsurface peat horizons [34,35]. The nature of the accumulation and migration of microelements in the STL is determined by the stability of the corresponding HM humates. The stability constants are arranged in the following row (pH = 3.7) Hg > Fe^3+^ > Al > Pb > Cu > Cr > Ni = Cd = Zn > Co = Mn [47]. A reliable correlation was established between gross mercury (*r* = 0.65, *n* = 29) and the mass fraction of HAs from Plot 2.

The intense accumulation of mercury falls on a layer of 20–30 cm. The highest ingress of mercury (about 80%) falls on the industrial period starting from the 17th century [18,48], which, based on radiocarbon dating data for the studied profiles, should correspond to the upper peat layer with a thickness of no more than 10–20 cm. It follows from the above that the mercury compounds formed at the place of deposition have to partially migrate down the profile. When combining the graphs of the vertical mercury distribution (Figure 5A) in the studied peatlands on soils with vegetation cover and on peat patches, it follows that a layer of 15–20 cm corresponding to the highest accumulation of the element was subject to erosion. The hypotheses of Scandinavian scientists indicate that eroded spots formed several hundred years ago [31]. The second and third phases of the Little Ice Age accounting for the minimum temperatures for the middle and late Holocene fall in the period of 16–17 centuries. It is likely to have caused the intensification of cryogenic processes, which led to the erosion of the surface layer of peat.

The nature of the depthwise distribution of the remaining elements has the following features: their minimum content in the oligotrophic and mesoeutrophic parts of the profile (to a depth of 50–70 cm); the accumulation at the boundary of the PL (cryogenic geochemical barrier); a systematic increase in the concentration of elements towards the central and lower parts of the profile with a maximum at a depth of 190–220 cm; and a lower concentration closer to the soil-forming material (except for Ni, Cr, Co, V, Fe, Al, Ba and Sr). The HM content in the eutrophic part of the profile is 3–200 times higher (especially Zn, Mn, Cd and As) than in the STL. The revealed anomalous concentrations of these elements in the central part of the peat stratum, which formed during the period of the Holocene optimum, cannot be explained by the fractionation of chemical elements from the atmosphere in the early periods of the Holocene.

Due to the low content of acidic complex-forming centers of HA molecules—salicylate, pyrocatechol, phthalic groups [49] in the upper layers of peat—most HMs have a low chemical affinity for HAs. Under conditions of increased acidity, HMs are highly mobile, migrating to the PL boundary. FA and low molecular weight organic acids, such as lactic, propionic, glycolic, as well as malic and hydroxybutyric acids, which are contained in large quantities in the STL [50], demonstrate high mobilizing ability through increasing the solubility of HM compounds.

Significant correlation coefficients for the content of Cd (*r* = 0.71, *r*_cr_ = 0.43, *n* = 21, *p* = 0.95), Zn (*r* = 0.64), Mn (*r* = 0.52), alkaline and alkaline-earth metals K (*r* = 0.54), Na (*r* = 0.65) and Mg (*r* = 0.70) with a mass fraction of FAs, indicate their chemical relationship with the labile fraction of macromolecular HAs. The STL displays a statistical correlation of Hg with the mass fraction of FAs (*r* = 0.44) and Cl^−^ ions in the water extract (*r* = 0.68). Under natural conditions, among the soluble inorganic forms that can exercise a significant effect on the migration characteristics of an element in the soil profile, stable chloride complexes are the most important [33,51]. In the STL, with a quantitative excess of chloride ions (Figure 2D) and a constant supply of high- and low-molecular weight organic acids during the decomposition of plant material, mercury and other elements can partially migrate to permafrost. The chemical analysis of the composition of elements in the water extract from peat shows some increase at the permafrost boundary, which is especially pronounced with regard to this indicator in relation to the content in the acid extract (*ω_E-H_*_2_*_O_/ω _E-acid_*) (Figure 1).

At the permafrost boundary, there is an increase in the proportion of water-soluble extract against the total content for a number of elements: Pb (up to 1.4%), V (up to 2.1%), Cr (up to 2.7%), Cd (up to 3.4%), Zn (up to 4.0%), Mn (up to 7.1%) and As (up to 15%), as well as alkaline (up to 55%) and alkaline-earth elements (up to 15%). Permafrost peat layers act as a cryogenic geochemical barrier to the migration of elements with a lower chemical affinity for HAs. Among all the studied elements, the proportion of water-soluble forms of Hg, Fe and Cu is below 0.5%, and for Pb, Al and Ni it is below 2%. These elements in peat deposits can be represented by stable humates, sulfides, (hydro) oxides, and silicates. The proportion of water-soluble As forms reaches a maximum of 20% in the central part of the profile and indicates that a significant part of As in the peat profile is represented in the form of soluble arsenite and arsenate ions, which are biologically available to peat-forming plants. The abnormal concentration of the element is associated with its increased regional background [20], and the biological accumulation of compounds of this element is illustrated by a fading decrease in its total content as you go from the lower layers towards the upper ones (Figure 5F). The increased content of water-soluble Mn compounds in the permafrost composition is also explained by a significant biological accumulation of the element and its mobility under reducing conditions [52,53].

The fractional composition of HMs in soils is characterized by an increased proportion of less strongly bound fractions as the technogenic impact increases [54]. This provision is confirmed by a rather high proportion of the water-soluble HM fraction in the subsurface part of the studied soils (especially Mn, As and Cd) (Figure 6).

The increased relative proportion of the water-soluble fraction, both slightly above and below the permafrost boundary, is indicative of the intersection of two independent processes there: the downward migration of elements with soil solutions down the profile (the modern process), and the upward biogenic migration of elements occurring normally as peat strata accumulated, which stopped 2000–2500 cal. year BP after the formation of permafrost [31] in the area under study. The second provision is explained by the availability of this HM fraction to plant organisms [31].

The nature of the depthwise distribution of water-soluble forms of elements and the data of correlation analysis show a significant correlation between the relationship of Zn, V, Cd, Cu and Pb (*r* = 0.60–0.87, *r*_cr_ = 0.43, *n* = 11) in the STL and sulfur content in the water extract, chloride ions (*r* = 0.60–0.93) with a mass fraction of total nitrogen (*r* = 0.60–0.97), and phosphorus with Mg, K, Na, Cd, Pb and Zn (*r* = 0.65–0.99); i.e., in the STL they are mainly presented in the form of sulfate, chloride, nitrate and phosphate ions. The STS show a statistical correlation between TOC and the content of alkaline and alkaline-earth elements (*r* = 0.71–0.97), and also the concentration of Cd, Zn, As and Pb (*r* = 0.60–0.99) in the water extract. This indicates the migration of elements as part of labile soil organic matter, which is formed along with the decomposition of plant residues under the acidic conditions of acrotelm. The permafrost composition displays a significant correlation between most macro- and microelements (except for Pb, Cr, As and Na) and sulfur content (*r* = 0.82–0.98, *r*_cr_ = 0.43, *n* = 21, *P* = 0,95), sodium is bound with Cl^−^ ions (*r* = 0.60), and As with dissolved organic matter (*r* = 0.77).

It is likely that under the anaerobic conditions of permafrost in peatlands, epigenetic hydrogen sulfide physicochemical barriers formed as a result of the activity of sulfate-reducing bacteria have to be considered as some of the most powerful ones [55]. A significant correlation of the total sulfur content with chalcophile elements Hg (*r* = 0.57, *r*_cr_ = 0.43, *n* = 21, *P* = 0.95), Pb (*r* = 0.42), Cd (*r* = 0.61), Cu (*r* = 0.66), As (*r* = 0.53), and other elements, Fe, Mn, Co and Ni (*r* = 0.47–0.67), is indicative of the formation of sulfides and the confinement of As as an admixture to sulfides, possibly pyrites. These barriers can be the reason for the abnormal accumulation of chalcophile elements (Hg, Cd, Pb, Ni, As and Zn), which are toxic to most organisms. It may explain the abnormal accumulation of As and Cd in the lower part of the studied sections. Pyrite crystals formed on the surface of plant residues and authigenic minerals inside plant cells may act as the most powerful concentrators of chalcophile elements [23].

It is probable that the accumulation of Fe and siderophile elements in the lower part of the profile is associated with the influence of aluminosilicate rocks. The transition to a gley circumneutral environment causes the emergence of two geochemical barriers—redox and sulfide [56]. Aluminum-iron complexes with HAs act as significant sorption geochemical barriers in the permafrost composition [53]. One can observe a high significant correlation of Fe and Al with siderophile elements: Ni, Cr, Mn, Co and V (*r* = 0.76–0.96, *r*_cr_ = 0.23, *n* = 43, *p* = 0,95), alkaline Na and K (*r* = 0.48–0.86) and alkaline-earth elements Mg, Sr and Ba (*r* = 0.87–0.96, *r*_cr_ = 0.23, *n* = 43, *p* = 0.95). Other HMs are also related to Al and Fe: Cu (*r* = 0.68–0.83, *r*_cr_ = 0.23, *n* = 43, *p* = 0.95), Pb (only with Fe, *r* = 0.53), Hg (*r* = 0.64–0.66), Cd (*r* = 0.73–0.87) and As (*r* = 0.64–0.84).

The significant accumulation of acid-soluble forms of Ca, Mg, and Fe in the lower part of the profile may be associated with a more intense development of microflora under anaerobic conditions, which results in a shift in the carbonate equilibrium and precipitation of poorly soluble Ca and Mg compounds [56]. Meanwhile, heteropolar salts of HAs with Ca^2+^ и Mg^2+^ ions may lead to the formation of bridges with clay minerals, which are powerful HM sorbents. A significant statistical correlation was revealed between calcium and most of the macro- (*r* = 0.57–0.64) and microelements (*r* = 0.56–0.98), especially Mn, Sr, Co, Ni and Fe.

A number of studies have shown a low sorption capacity of oligotrophic types of peat with regard to some HMs, and a significant concentration of elements in eutrophic peat types [4,57]. A statistical analysis of the correlation between the microelement content in stratified peat horizons and the composition of peat formers was carried out based on the botanical composition data. The plant material of the STL peat is dominated by green and sphagnum mosses, subshrubs, and vascular plants. Numerous studies indicate the use of bryophytes as bioindicators of atmospheric metal deposition, since they mainly absorb elements from the atmosphere [58,59]. Mosses can accumulate metals in excess of their physiological needs for the morphological structure of cells and tissues [60]. Hg, Cd, Pb and Zn show significant affinity for the green mosses *Polytrichum* sp., *Dicranum* sp. (*r* = 0.46–0.66) and sphagnum mosses *Sphágnum Subsecunda*, sp. *squarrosum*, sp. *russowii*, sp. *riparium* (*r* = 0.49–0.97), as well as for alkaline and alkaline-earth elements (*r* = 0.58–0.92). The analysis of the bryophyte composition reveals that such toxic elements as Hg, Pb and Cd are emitted in significant quantities, not only in various types of production but also during coal combustion [61,62,63]. In our opinion, hummocky peatlands act as a more effective indicator of atmospheric pollution, if compared to typical oligotrophic bogs, due to the high degree of peat decomposition and the content of the main HM–HA binding sites.

In addition to mosses, vascular plants also displayed a significant accumulation of HMs in the studied areas. We have revealed a significant correlation between the content of heather subshrubs in the composition of the STL peat and the mass fraction of Hg (*r* = 0.79, *r*_cr_
*=* 0.43, *n* = 21), Pb (*r* = 0.65), Cd (*r* = 0.50) and Zn (*r* = 0.50). Copper is accumulated in *Scheuchzeria* (*r* = 0.79), in *Eriophorum* (*r* = 0.75), in *Menyanthes trifoliata* (*r* = 0.57) and in *Carex limosa* (*r* = 0.52). Siderophile elements (Fe, Ni, Cr, V, Co) have the strongest statistical correlation with *Eriophorum* (*r* = 0.43–0.75), *Scheuchzeria* (*r* = 0.63–0.86) and *Menyanthes trifoliata* (*r* = 0.44–0.69). Regarding sedge species, *Carex rotundata* are bound to Ni, Co, Mn and V (*r* = 0.46–0.74), *Carex limosa* to Ni, Co and V (*r* = 0.46–0.61), *Salix* shrubs are statistically related to V and Mn (*r* = 0.50–0.58), and Fe and Cr to woody *Betula pubescens* (*r* = 0.56–0.61).

At the early stage of peatland formation under eutrophic conditions, the biogenic factor of element accumulation is of utmost importance. Since peat formation is an accumulative process, the nutrients move upward in a biogenic way in the growing peat deposit at the early development stage [53]. The accumulation pattern in the lower part of the profile is conditioned by the composition of groundwater and soil-forming material, and the specific accumulation of mineral components by “biological pumps”. Certain plant species, such as *Carex cespitosa* (*r* = 0.48–0.77, *r*_cr_ = 0.29, *n* = 43, *p* = 0.95) and *Equisetum* (*r* = 0.32–0.77), which have a significant statistical correlation with, unexceptionally, all elements in the permafrost peat layer of hummocky peatlands in Plots 1 and 2, may serve to be such “pumps”. Sedges and horsetails are typical representatives of eutrophic plant communities; they are among the most high-ash species that effectively absorb compounds of Al, Ca, Mg, Na, Fe, Mn and other metals, as well as silicic acid. Additionally, most of the elements are accumulated by woody *Betula pubescens* species and *Salix* shrubs [64].

The overlay of the graphs of the depthwise element distribution of the studied sections shows lower levels of HM accumulation in the eutrophic part of the Plot 3 profile. It may be associated with both the hydrochemical features of the studied wetland system in the early periods of the peat deposit formation, and with the mineral composition of the soil-forming material. However, the botanical composition analysis of these sections revealed a significantly lower proportion of species referred to as powerful biological concentrators, namely *Equisetum,* and the complete absence of *Carex cespitosa* in the paleovegetation composition. This indirectly confirms the hypothesis of classifying these genera and plant species as “biological pumps”.

## 5. Principal Component Analysis

The content of toxicants, botanical composition, physical and chemical parameters of peat samples (Plots 1 and 2) from the STL and from the PL were further subjected to statistical analysis using the PCA method (Figure 7).

The results of the PCA for peat samples from the STL explained 75.36% of the total variability. The dimension of the 31 input variables was reduced by PCA to two principal components with eigenvalues higher than two: the first axis (PC1) explained 50.67%, and the second (PC2) 24.70% of the total variability, while the third axis (PC3) explained 7.27%. The PC1 was positively associated with the content of Pb, Hg, Cd, Zn and the share of subshrub and mosses, and negatively coordinated with the content of Cu, Cr, Ni, Co, V, Al, Ba and the share of herbs, TOC and pH on this axis in peat soils. The positive correlation of the factor is associated with the accumulation of most chalcophile elements in the upper part of the profile due to their absorption by mosses, while the negative one is explained by the migration of most siderophile elements with water-soluble organic components (FAs, low molecular weight acids). PC2 values indicate the effect of higher pH values on the accumulation of Mn, As, Ca and Sr in the STL.

The results of the PCA for peat samples from the PL explained 71.04% of the total variability. The first axis (PC1) explained 56.75%, and the second (PC2) 14.29% of the total variability, while the third axis (PC3) explained 9.34%. The PC1 was negatively associated with the content of most chemical elements, causing their accumulation on aluminum-iron minerals. The positive correlation of PC2 with S, As, Cd, Cu, Pb and Hg is associated with their interaction with the hydrogen sulfide geochemical barrier.

## 6. Conclusions

Data on the content and composition of acid- and water-soluble forms of HMs (Hg, Pb, Cd, Cu, Zn, Cr, V, Ni, Mn, Co, Sr and Ba) and As were obtained in the stratified horizons of hummocky peatlands of background and technogenic landscapes in the extreme northern taiga of the European Northeast. Comparisons of data on the HM content in the upper horizon of the contaminated soil with the background ones revealed an excess of chalcophile elements—Hg, Cd, Pb, Cu and As—and siderophile elements: Ni, Co, Cr and V. Spheroidal microparticles (from <1 μm to 30–50 μm in size), which appear to be aluminosilicates, often with the inclusion of iron oxides (hydroxides) and an admixture of other metals, were found in the upper layer of peat in Plot 3. The presence, composition and morphology of these particles made it possible to use them as a marker of the area pollution by products of high-temperature coal combustion.

The effect of geochemical (sorption, cryogenic, biogenic, redox) barriers on the intraprofile distribution of HM and As has been established. A significant biogenic upward migration of most elements is shown to be carried out by woody-sedge and sedge communities under eutrophic conditions. The conducted correlation analysis revealed the accumulation of HMs by green (*Polytrichum* sp., *Pleurozium* sp., *Dicranum* sp.) and sphagnum (*Sphagnum russowii*, sp. *Subsecunda*, sp. *squarrosum*) mosses, and subshrubs (*Ericales, Ledum*) in the STL. The accumulation of Hg and Pb in the STL of the studied peatlands is explained by the formation of stable coordination compounds with HAs (a sorption geochemical barrier). The high acidity of peat in the STS intensifies the mobility of most microelements. Under modern conditions, the upper boundary of permafrost serves as a geochemical barrier for the downward migration of elements with low chemical affinity for HAs. The results of the PCA analysis revealed that the accumulation of the studied toxicants in the PL is associated with their sorption on aluminum–iron minerals and interaction with the hydrogen sulfide barrier.

Thus, with the current trend towards climate warming, the permafrost thawing of peatlands in the cryolitic zone poses a real environmental threat of the influx of inorganic toxicants to the hydrological network and their inclusion in food chains. Data on the background content of HMs and As can be used in the environmental monitoring of Histosols.

## Figures and Tables

**Figure 1 ijerph-20-03847-f001:**
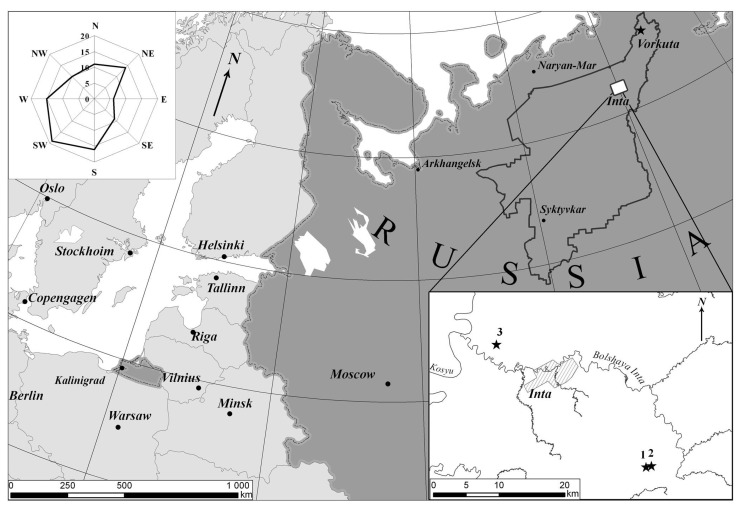
Location of sampling points of conditionally background plots (Plots 1 (**1**) and 2 (**2**)) and under the influence of aerotechnogenic pollution (Plot 3 (**3**)).

**Figure 2 ijerph-20-03847-f002:**
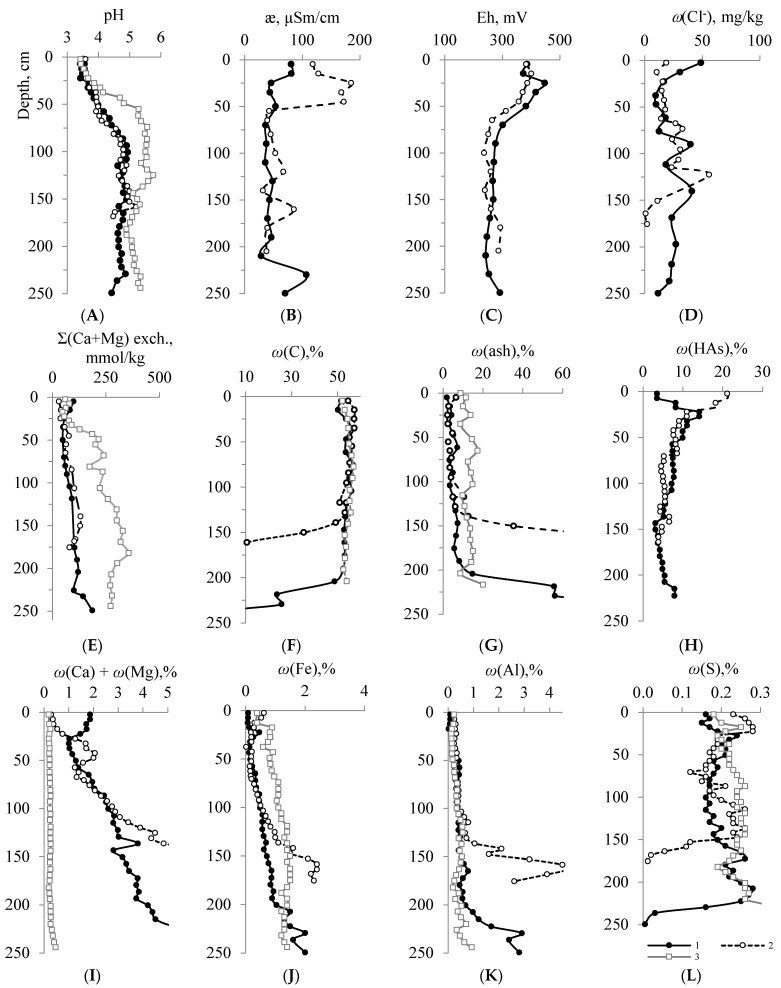
Along-profile variations of water extracts: pH (**A**), the electric conductivity (**B**), redox potential (**C**), Cl^−^ ions in water extracts (**D**), total of exchangeable Ca and Mg (mmol/kg) (**E**), total carbon content (**F**), mass fraction of ash (**G**), mass fraction of HAs (**H**), mass fraction of acid-soluble forms of elements: total of Ca and Mg (**I**), Fe (**J**), Al (**K**) and S (**L**) in peat samples from Plots 1 (1), 2 (2) and 3 (3).

**Figure 3 ijerph-20-03847-f003:**
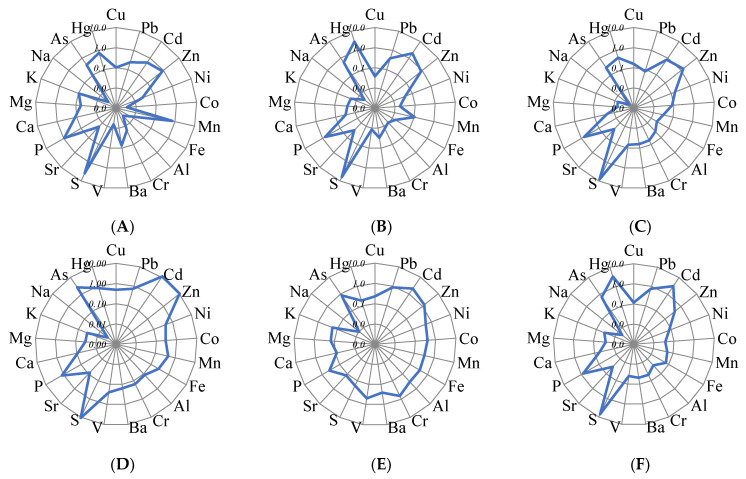
Concentration factors of elements relative to their abundances in the Earth’s crust (logarithmic scale) in Plot 1 (conditional background): plant material (**A**), hor. H1 0–5 cm (**B**), hor. H5 40–50 cm (upper frozen horizon) (**C**), hor. H16 205–211 cm (**D**), hor. Cg 240–259 cm (parent rock) (**E**), and in Plot 3 (local aerotechnogenic pollution): hor. H1 0–5 cm (**F**).

**Figure 4 ijerph-20-03847-f004:**
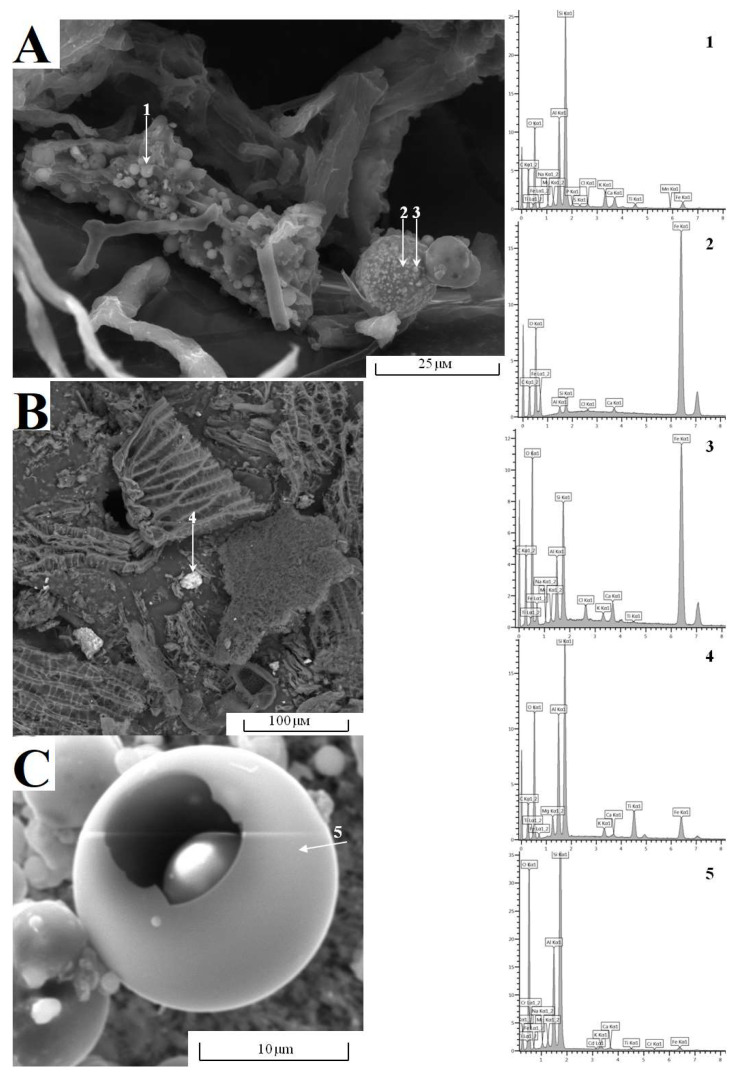
Photomicrographs of peat samples from hor. H1 0–5 cm from Plot 3 (local aerotechnogenic pollution) (**A**) and hor. H1 0 to 5 cm from Plot 1 (conditional background) (**B**), plerosphere example from hor. H1 0 to 5 cm from Plot 3 (**C**), and the results of the elemental analysis of particles of different origin (1–5).

**Figure 5 ijerph-20-03847-f005:**
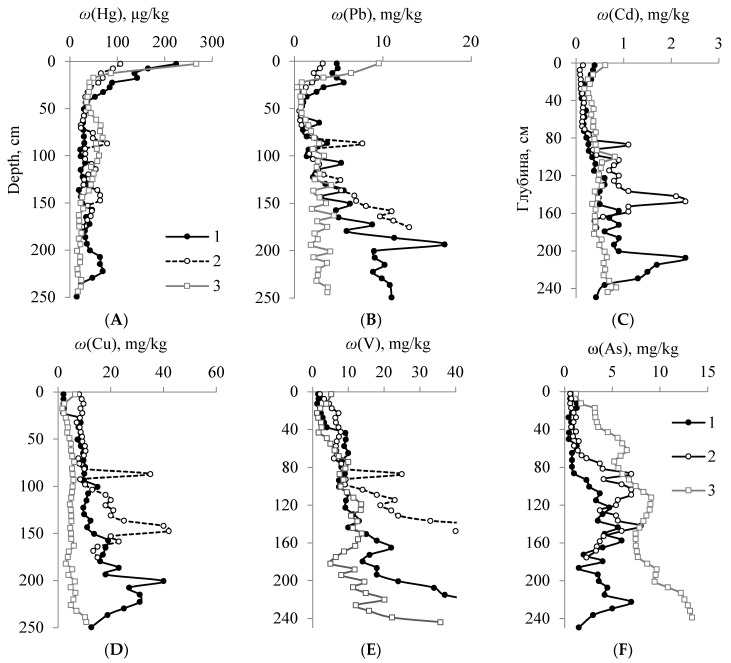
Along-profile variations of total Hg (**A**) and acid-soluble forms of Pb (**B**), Cd (**C**), Cu (**D**), V (**E**) and As (**F**) in peat samples from Plot 1 (1), Plot 2 (2) and Plot 3 (3).

**Figure 6 ijerph-20-03847-f006:**
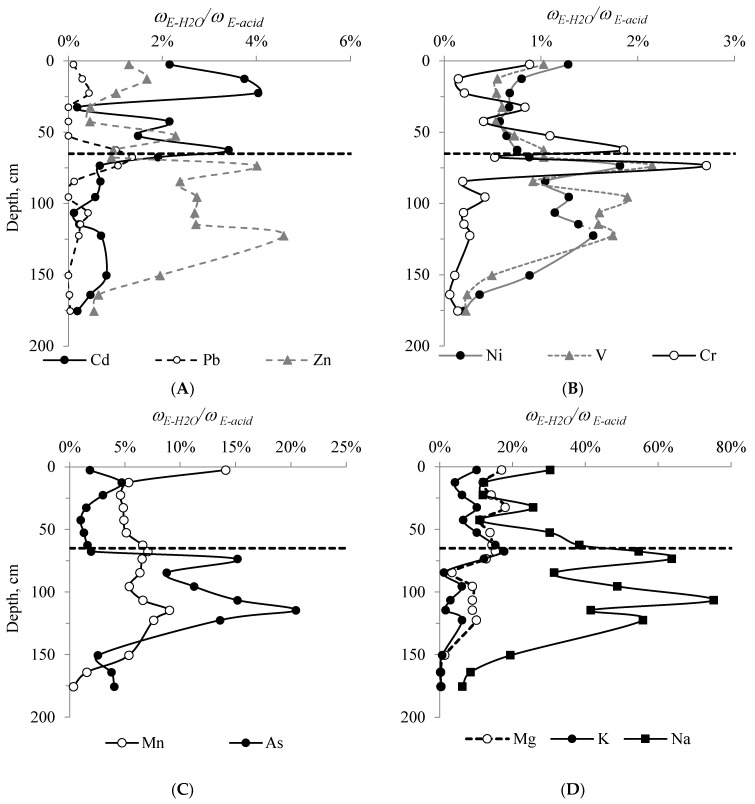
The proportion of water-soluble forms of elements in relation to acid-soluble form: Cd, Pb, Zn (**A**); Ni, V, Cr (**B**); Mn, As (**C**); Mg, K, Na (**D**) in peat samples from Plot 2.

**Figure 7 ijerph-20-03847-f007:**
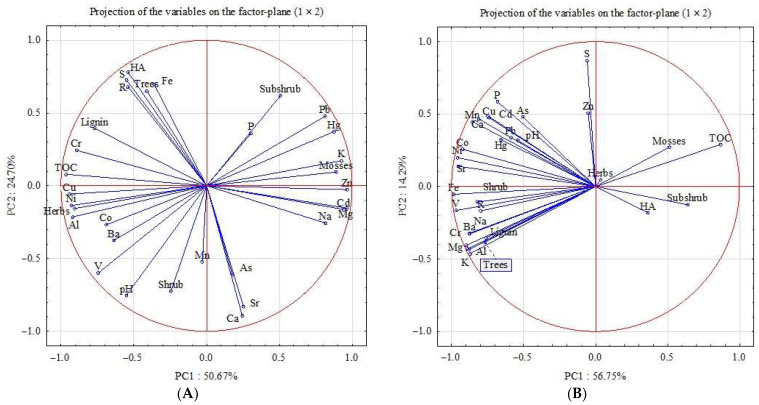
Projection of peat composition parameters using PCA: STL (**A**) and PL (**B**).

**Table 1 ijerph-20-03847-t001:** Chronology and dominant botanical species of peat layers.

Depth, cm	Peat Layer	Peat Type	*R*,%	Chronology		Dominant Botanical Species
				^14^C-Age (Year BP)	Calibrated Age (Cal Year BP)	
Plot 1. Fibric Folic Cryic Histosol						
0–10	H1	Raised	20	n.d.	27	*Sphagnum, Ericales, Pleurozium*
10–20	H2	Raised	15	n.d.	n.d.	*Sphagnum, Polytrichum, Ericales*
20–30	H3	Raised	20–25	2080 ± 60	2054	*Sphagnum*
30–40	H4	Transitional	25	n.d.	n.d.	*Scheuchzeria, Sphagnum*
40–54	H5	Transitional, frozen	25–30	n.d.	n.d.	*Scheuchzeria, Carex, Eriophorum*
54–69	H6	Transitional, frozen	30	n.d.	n.d.	*Scheuchzeria, Carex, Eriophorum*
69–83	H7	Transitional, frozen	25–30	n.d.	n.d.	*Sphagnum, Scheuchzeria, Carex, Eriophorum, Equisetum*
83–97	H8	Transitional, frozen	25–30	n.d.	n.d.	*Sphagnum*, *Scheuchzeria, Carex, Eriophorum, Menyanthes*
97–111	H9	Transitional, frozen	25–30	n.d.	n.d.	*Sphagnum*, *Scheuchzeria, Carex, Eriophorum, Menyanthes*
111–126	H10	Fen, frozen	25–30	n.d.	n.d.	*Sphagnum, Warnstorfia, Scheuchzeria, Carex, Eriophorum, Equisetum*
126–140	H11	Fen, frozen	35	n.d.	n.d.	*Sphagnum, Warnstorfia, Scheuchzeria, Carex, Eriophorum, Equisetum*
140–154	H12	Fen, frozen	30–35	n.d.	n.d.	*Sphagnum*, *Carex, Eriophorum, Equisetum, Menyanthes*
154–169	H13	Fen, frozen	30–35	n.d.	n.d.	*Sphagnum*, *Carex, Equisetum, Menyanthes*
169–183	H14	Fen, frozen	30–35	n.d.	n.d.	*Sphagnum*, *Carex, Equisetum, Menyanthes*
183–197	H15	Fen, frozen	30–35	n.d.	n.d.	*Sphagnum*, *Carex, Equisetum, Menyanthes*
197–211	H16	Fen, frozen	30–35	n.d.	n.d.	*Carex, Eriophorum, Equisetum, Menyanthes*
211–226	H17	Fen, frozen	30–35	n.d.	n.d.	*Carex, Equisetum, Menyanthes*
Plot 2. Hemic Folic Cryic Histosol (Turbic)						
0–10	H1	Raised	>50	2680 ± 70	2804	*Eriophorum, Scheuchzeria, Betula pubescens, Ericales*
10–20	H2	Raised	30–35	2570 ± 60	2635	*Scheuchzeria, Eriophorum, Betula pub., Ericales*
20–30	H3	Transitional	30	n.d.	n.d.	*Scheuchzeria, Carex*
30–40	H4	Transitional	30	4640 ± 70	5388	*Scheuchzeria, Carex, Eriophorum*
40–50	H5	Transitional	25	n.d.	n.d.	*Scheuchzeria, Eriophorum*
50–60	H6	Transitional	25	n.d.	n.d.	*Scheuchzeria, Eriophorum, Carex*
60–70	H7	Transitional	25	4920 ± 70	5661	*Scheuchzeria, Carex, Eriophorum*
70–79	H8	Transitional, frozen	25	n.d.	n.d.	*Scheuchzeria, Carex, Eriophorum*
79–90	H9	Transitional, frozen	25	5980 ± 80	6823	*Scheuchzeria, Carex*
90–101	H10	Transitional, frozen	25–30	n.d.	n.d.	*Scheuchzeria, Carex*
101–112	H11	Fen, frozen	30–35	6510 ± 90	7421	*Menyanthes, Equisetum, Carex*
112–123	H12	Fen, frozen	30	n.d.	n.d.	*Menyanthes, Equisetum, Carex*
123–134	H13	Fen, frozen	25–30	n.d.	n.d.	*Sphagnum, Equisetum, Carex*
134–145	H14	Fen, frozen	30–35	7010 ± 90	7839	*Carex cespitosa, Equisetum*
145–156	H15	Fen, frozen	35–40	n.d.	n.d.	*Carex cespitosa, Betula pub., Equisetum*
156–167	H16	Fen, frozen	45–50	8060 ± 180	8953	*Carex ces., Equisetum, Betula pub.*
Plot 3. Fibric Folic Cryic Histosol						
0–10	H1	Raised	20–25	n.d.	n.d.	*Dicranum, Ledum, Betula nana, Ericales*
10–20	H2	Transitional	30	n.d.	n.d.	*Betula* sp., *Ericales, Ledum*
20–30	H3	Fen	30	n.d.	n.d.	*Carex, Betula* sp., *Menyanthes*
30–40	H4	Fen	25–30	n.d.	n.d.	*Carex, Betula* sp., *Menyanthes*
40–52	H5	Fen, frozen	30	n.d.	n.d.	*Carex, Betula* sp., *Menyanthes*
52–65	H6	Fen, frozen	25	n.d.	n.d.	*Carex, Menyanthes, Betula* sp.
65–77	H7	Fen, frozen	35	n.d.	n.d.	*Carex, Betula* sp., *Salix*
77–90	H8	Fen, frozen	35	n.d.	n.d.	*Carex, Betula* sp., *Salix*
90–103	H9	Fen, frozen	35–40	n.d.	n.d.	*Carex, Betula* sp., *Salix*
103–115	H10	Fen, frozen	35–40	n.d.	n.d.	*Carex, Equisetum, Menyanthes*
115–128	H11	Fen, frozen	35–40	n.d.	n.d.	*Carex, Equisetum, Menyanthes*
128–141	H12	Fen, frozen	35–40	n.d.	n.d.	*Carex, Equisetum*
141–153	H13	Fen, frozen	35–40	n.d.	n.d.	*Carex, Equisetum*
153–166	H14	Fen, frozen	35–40	n.d.	n.d.	*Carex, Betula* sp., *Salix, Equisetum*
166–179	H15	Fen, frozen	35–40	n.d.	n.d.	*Carex, Equisetum, Calliergon*
179–191	H16	Fen, frozen	30–35	n.d.	n.d.	*Carex, Equisetum, Calliergon*
191–204	H17	Fen, frozen	30–35	n.d.	n.d.	*Carex, Calliergon, Equisetum*
204–217	H18	Fen, frozen	25–30	n.d.	n.d.	*Carex, Menyanthes, Equisetum*

*R*—degree of peat decomposition; n.d.—not determined.

## Data Availability

Not applicable.

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
