# Peer review of "Geochemical Characteristics of the Vertical Distribution of Heavy Metals in the Hummocky Peatlands of the Cryolithozone"

_ijerph, 2023, doi:10.3390/ijerph20053847_

Round 1
Reviewer 1 Report
Journal: IJERPH (ISSN 1660-4601) Environmental Research and Public Health
Manuscript ID: ijerph-2233779
Type: Article
Title: Geochemical Characteristics of the Vertical Distribution of Heavy Metals in the Hummocky Peatlands of the Cryolithozone
Authors: Roman Vasilevich1*, Mariya Vasilevich1, Evgeny Lodygin1 and Evgeny Abakumov
Section: Environmental Earth Science and Medical Geology
Special Issue: Heavy Metals in Soil: Source, Transformation, Transfer, Risk and Pollution Remediation
Comments for Authors:
1.- Abstract: The abstract is very long and theoretical in the initial part (Lines 10 to 15); this could be part of the Introduction Section.
The authors state three objectives (Lines 16 to 20), however, these are very broad and ambiguous, and need to be made more concrete.
The authors do not mention the methods that will be applied to meet the stated objectives. It is recommended to mention them briefly and concisely.
Lines 31 to 33. The authors make a predictive approach without a firm justification related to this work. Please clarify.
It is recommended that the authors add a paragraph at the end of the Abstract in which they justify how the results obtained could be used in a practical way.
2.- Keywords: authors are encouraged to include keywords in the abstract.
3.- Introduction (Lines 76 to 80): The text that appears in the Abstract (see Lines 16 to 20) is repeated verbatim. It should be rewritten.
Authors must highlight the novelty of this work at the end of the Introduction. Fix.
4.- Section 2.2. Field Sampling. It is recommended that the authors rename this Section as "Materials".
On the other hand, the arguments written in Lines 82 to 92, as well as Lines 102 to 109, should be deleted.
5.- Figure 1. The location of the sampling points is not clear. The authors use two maps, but it is difficult to relate them. Fix.
6.- Figure 4. EDS (1 - 5) have poor resolution. It is recommended to improve it.
7. - Line 286. The phrase "Nanoparticles also occur." appears in isolation and does not seem to make sense. Fix.
On the other hand, elements are missing in the sentence "The X-ray phase analysis". Fix
8.- Line 289. The way the word "(hydro)oxides" is written is very confusing. Fix.
9.- Section 4. Results and discussion. The authors give a comprehensive presentation of the results obtained, but the discussion of the results is rather poor and does not compare in detail with the work of other researchers. It would be advisable to introduce more references in this part of the paper in order to establish how far this research has progressed.
10.- The similarity index is very high, although the coincidences found in chapters 3 and 4.3.3 are of particular concern. It is recommended that this index be lowered in order to be published.
Author Response
The authors thank the reviewer for carefully reading our manuscript. We have corrected all comments as far as possible. All changes are highlighted in yellow.
1. Abstract:
We have shortened the introduction section. Added analysis methods. Some suggestions have been corrected. Due to the fact that the volume of the abstract is limited, we added information about the use of the results obtained in practice in conclusion.
2. Keywords are included in the abstract
3. Usually, the formulated goals and objectives are the same in the abstract and in the introduction. But following your remark, we expanded and concretized the tasks in the text of the article, and left an abbreviated version in the abstract. We emphasized the novelty of our research.
4. Section 2 renamed as "Materials". In our opinion, climate and land cover data are important for describing the study area and general understanding of the article.
5. Fig. 1 has been edited.
6. Resolution is improved in Fig. 4.
7. Phrase deleted. The sentence "The X-ray phase analysis" is taken from (Kotova et. Al, 2015). The link was provided earlier.
8. the word "(hydro)oxides" corrected in text
9. The discussion has been expanded, and new relevant references have been added.
10. Yes, the similarity index was somewhat high in chapters 3 and 4.3.3. and the editorial board recommended that the text of these chapters be revised. We corrected these chapters in the next version of the manuscript, and also explained to the editors that section 3 "Methods" can be difficult to describe in different words, since the description of methods includes a large number of standard phrases. The editors accepted our correction and agreed with our arguments.
Reviewer 2 Report
Wetland ecosystems are one of the main reservoirs depositing various classes of pollutants in high-latitude regions. Climate warming results in the degradation of permafrost in cryolitic peatlands, which exposes the hydrological network to risks of heavy metal (HM) ingress and its subsequent migration to the Arctic Ocean basin. This study aims to analyze the content of acid- and water-soluble forms of HM in peat soils. The study focused on the characteristics of the layer-by-layer accumulation of HMs and As in hummocky peatlands of the extreme northern taiga of the European Northeast. It revealed that the upper level of microelement accumulation is associated with the STL due to aerogenic pollution. Specifically composed spheroidal microparticles found in the upper layer of peat may serve as indicators of the area polluted by power plants. The accumulation of water-soluble forms of most of the pollutants studied on the upper boundary of the permafrost layer (PL) is explained by the high mobility of elements in an acidic environment. The following comments need to be addressed by the authors.
Define the abbreviation “STL” in the abstract at his first appearance. “antropogenic impact” or “anthropogenic impact”?. impacThe manuscript lacks a review of recent literature (2021, 2022 and 2023). Define the Research gap identification from the recent and relevant literature. State the main contribution of the present study. The manuscript's structure needs to be mentioned at the end of the introduction section. Avoid clustered references, e.g. [5-7], [13-15], [16-18]. Provide the significant findings of each reference in the text as much as possible. Remove the obsolete and old references. Provide a high-resolution for Figure 1. The right-side figures of Figure 4 have very poor clarity and are not readable. Provide high-resolution images. The measurement uncertainty is to be estimated and provided. Improve the picture quality of Figures 5 to 7. English and grammar must be improved.
Author Response
The authors thank the reviewer for carefully reading our manuscript. We have corrected all comments as far as possible. All changes are highlighted in yellow.
1. Abbreviation “STL” was added to the abstract.
2. “anthropogenic impact” changed to “anthropogenic impact”
3. We have replaced a number of old references with new ones (marked in yellow).
4. The quality of figures has been improved
5. English and grammar have been additionally checked
Reviewer 3 Report
This study first carryed out a quantitative analysis of the content of acid- and water-soluble 16 forms of HM and As across the profile of peat soils in background and technogenic landscapes, evaluating the contribution of the antropogenic impact to the accumulation of trace elements in the STL of peat deposits, and then discoverred the effect of biogeochemical barriers on the vertical distribution of HMs and As. The study focused on the characteristics of the layer-by-layer accumulation of HMs and As in hummocky peatlands of the extreme northern taiga of the European Northeast.
Although there are three samples, some results have been achieved. I suggest authors revising the MS in detail list as follows.
line 93, 96, should be revised as (65° 54 ΄05.0˝N, 60° 26΄ 93 34.5˝E),( 66° 05 ΄05.4˝N, 59° 58΄ 36.3˝E )
Fig.1 Missing scale bar and north compass in map.
line 116 what is the unit of 50 x 50?
line179 what is the R ?
line 258 what is the SEM?
the submap in Fig.4 is not clear.
Author Response
The authors thank the reviewer for carefully reading our manuscript. We have corrected all comments as far as possible. All changes are highlighted in yellow.
1. line 93, 96 have been corrected (marked in yellow)
2. Fig 1 has been corrected
3. 50 x 50 changed to 50×50 cm
4. line 134 clarified peat decomposition degree (R)
5. see line 131: scanning electron microscope (SEM)
6. Figure 4 quality was improved
Reviewer 4 Report
The distribution of heavy metals in the hummocky peat soil profile was studied. The contribution of anthropogenic activities to the accumulation of heavy metals has been speculated. The results have important implications for the permafrost thawing that may release heavy metals into the water systems. However, I felt the connection between the accumulation of heavy metals and anthropogenic impacts needs more attention.
To have a better understanding of anthropogenic impacts on the accumulation of heavy metals in the peat soil profile, more plots may be needed between plots 1 (or 2) and 3, to see the anthropogenic input of heavy metals to the soil profile as a function of distance from the pollution source.
L189-91, “The top part of the STL (0–40 см) of peatlands shows a trend to increasing values of the specific electrical conductivity of the water extract up to 170–185 μSm/cm”. This is true only for plot 2, but not true for all plots.
L219, 224, 225, and many other places, the citation of figures is incorrect.
L227, define the total Clarke value.
Figure 2, plots 1 and 2 have similar variation trends in these parameters but in different depth ranges. Are these differences worthy of discussion?
Section 4.3.2, the identification of particle phases is vague. Was it completed by XRD? Identifying phases only based on elemental analysis like in Figure 4 is not accurate.
Figure 4, label SEM panels. Improve the resolution of images of elemental analysis 1-5.
Author Response
The authors thank the reviewer for carefully reading our manuscript. We have corrected all comments as far as possible. All changes are highlighted in yellow.
1. Unfortunately, there are no Histosols between plots 1 (or 2) and 3, so it is not possible to study intermediate plots.
2. The sentence in the text has been corrected for: «The top part of the STL (0–40 см) of peatlands shows a trend to increasing values of the specific electrical conductivity of the water extract up to 80 μSm/cm (Plot 1) and 170–185 μSm/cm (Plot 2)»
3. The numbering of figures in L219, 224, 225 and L233 has been corrected
4. L227 has been explained: «The values of the total concentration factor of the HMs relative to the average content in the earth's crust (ΣKi)…».
5. Explanation was given earlier in L362-365 using the distribution of mercury as an example.
6. An explanation of the method used has been added to section 3 "Methods": «The morphological analysis of mineral particles was determined on a Tescan Vega 3 LMH (Tescan, Czech Republic, Brno) scanning electron microscope (SEM). Phase analysis and elemental composition were performed by energy dispersive X-ray spectroscopy (EDS) using X-MAX 50 (Oxford instruments, Great Britain, Abingdon) at the Institute of Geology (Komi RC, Ur. Br., RAS)».
7. Figure 4 quality was improved
Round 2
Reviewer 1 Report
Thank you very much for the modifications made and that would justify the publication of this paper.
Best regards
Author Response
We thank the reviewer. Best regards, authors.
Reviewer 2 Report
Comments are addressed appropriately.
Author Response

(The authors gave the same response as above.)

Reviewer 4 Report
EDS cannot identify mineral phases but can estimate the chemical compositions of minerals. Suggest checking wording in the Methods and Results & Discussion sections.
Author Response
We thank the reviewer. We have edited the Methods section (line 131). Best regards, authors.